# Genomic characterization of malignant progression in neoplastic pancreatic cysts

Michaël Noë 📖 et al.#

Intraductal papillary mucinous neoplasms (IPMNs) and mucinous cystic neoplasms (MCNs) are non-invasive neoplasms that are often observed in association with invasive pancreatic cancers, but their origins and evolutionary relationships are poorly understood. In this study, we analyze 148 samples from IPMNs, MCNs, and small associated invasive carcinomas from 18 patients using whole exome or targeted sequencing. Using evolutionary analyses, we establish that both IPMNs and MCNs are direct precursors to pancreatic cancer. Mutations in *SMAD4* and *TGFBR2* are frequently restricted to invasive carcinoma, while *RNF43* alterations are largely in non-invasive lesions. Genomic analyses suggest an average window of over three years between the development of high-grade dysplasia and pancreatic cancer. Taken together, these data establish non-invasive IPMNs and MCNs as origins of invasive pancreatic cancer, identifying potential drivers of invasion, highlighting the complex clonal dynamics prior to malignant transformation, and providing opportunities for early detection and intervention.

---

#A list of authors and their affiliations appears at the end of the paper.

Pancreatic cancer is a deadly disease with a dismal prognosis that is predicted to soon be the second leading cause of cancer death in the United States[1]. However, like other epithelial malignancies, pancreatic cancer arises from noninvasive precancerous lesions that are curable if detected and treated early enough. Although the majority of pancreatic cancers are believed to originate in microscopic precancerous lesions (pancreatic intraepithelial neoplasia or PanIN), a significant minority arise in association with larger cystic neoplasms that can be detected using currently available imaging technologies[2]. These neoplasms, which include intraductal papillary mucinous neoplasms (IPMNs) and mucinous cystic neoplasms (MCNs), are frequently diagnosed incidentally on abdominal imaging, identifying a cohort of at-risk patients with an important opportunity for prevention of invasive pancreatic cancer[2]. However, prevention must be balanced with potential overtreatment of low-risk lesions, as pancreatic resection carries significant morbidity and even occasional mortality[3]. There is a critical need to understand the molecular alterations that are associated with the development of invasive cancer, as these represent potential biomarkers to identify cysts at high risk for progression to carcinoma and thus requiring clinical intervention.

Although genomic analyses have been performed on hundreds of invasive pancreatic cancers, relatively few noninvasive neoplasms have been analyzed comprehensively. Whole exome and targeted sequencing of small cohorts of IPMNs and MCNs have revealed driver genes characteristic of each type of cystic neoplasm[4–6], while targeted analyses in larger cohorts have confirmed the prevalence of specific driver gene mutations that correlate with grade of dysplasia or histological subtype[7]. These studies have confirmed that hotspot mutations in the oncogenes *KRAS* and *GNAS* occur in low-grade lesions while mutations in other driver genes, including *CDKN2A*, *TP53*, *RNF43*, and *SMAD4*, occur with increasing prevalence in lesions with high-grade dysplasia or associated invasive carcinoma[8]. Targeted next generation sequencing has been used to analyze pancreatic driver genes in different regions of IPMNs, revealing a surprising degree of intratumoral genetic heterogeneity, even with respect to well-characterized driver gene mutations[9–12]. However, the above analyses were based on studies of either single regions from each neoplasm or a limited number of genes from multiple regions, and did not provide an analysis of the evolutionary relationship between different regions of pancreatic cysts and associated cancers. These limitations highlight the need for comprehensive genomic analysis of these cysts and associated invasive cancers to understand the molecular alterations that underlie the transition to invasive carcinoma.

In this study we perform whole exome sequencing of IPMNs and MCNs and their associated invasive carcinomas. Importantly, we focus our study on small invasive carcinomas (<2.5 cm) in order to more precisely analyze the genetic alterations that occur at malignant transformation in pancreatic tumorigenesis. In addition, in a subset of our samples, we perform deep targeted next generation sequencing on a larger set of additional tissue samples in order to assess mutated loci through entire neoplasms, including areas of low-grade dysplasia, high-grade dysplasia, and invasive carcinoma. These analyses reveal important features of pancreatic tumorigenesis, including evolutionary relationships between different regions within cystic neoplasms as well as molecular alterations that may drive the transition from a non-invasive precursor lesion to invasive cancer.

## Results

**Overall approach**. In order to dissect the molecular relationships between non-invasive dysplastic lesions and invasive pancreatic cancers, we performed whole exome sequencing of 39 neoplastic tissue samples from 18 patients with small invasive carcinomas (<2.5 cm) associated with neoplastic pancreatic cysts, including 16 patients with IPMNs and 2 patients with MCNs (Supplementary Data 1). Whole exome sequencing was performed on one sample from the noninvasive component with high-grade dysplasia and one sample from the invasive cancer in each case, and for three cases an additional noninvasive sample with low-grade dysplasia was also analyzed by whole exome sequencing. Matched normal samples were analyzed by whole exome sequencing in each case to exclude germline variants and to identify somatic mutations. Whole exome sequencing was performed with an average total coverage of 177× (distinct coverage of 145×), generating 1.3 TB of sequencing data (Supplementary Data 2).

In addition to whole exome analyses, we performed targeted next generation sequencing of 109 microdissected tissue samples from seven of the above cases (six IPMNs and one MCN). For these targeted analyses, we performed laser capture microdissection to isolate neoplastic cells from every available tissue block of the noninvasive cyst and cancer specimens. Separate samples were microdissected based on grade of dysplasia, cell morphology, architecture, and spatial location. This resulted in 8–22 additional samples per case. The targeted panel analyses included all mutated loci identified in the whole exome sequencing of these seven cases, as well as the entire coding regions of 15 well-characterized pancreatic driver genes (Supplementary Data 3). The targeted sequencing had an average coverage of 508× (distinct coverage of 460×) (Supplementary Data 2).

We developed an integrated mutation calling pipeline to rigorously assess mutations in all sample types in our analyses in order to confidently identify even subclonal alterations in samples with low neoplastic purity (see "Methods") (Fig. 1, Supplementary Data 4). In addition, we utilized both on target and off target reads to examine focal copy number changes as well as loss of heterozygosity throughout the genome (Fig. 2, Supplementary Data 5 and 6). From our whole exome sequencing analyses, we identified an average of 66 somatic mutations in samples from noninvasive components (range 26–111) and an average of 65 somatic mutations in invasive carcinoma samples (range 31–105) (Fig. 1a, Supplementary Data 4). An average of 47 somatic mutations were shared between the noninvasive and invasive components, while 19 somatic mutations were unique to samples from noninvasive components and 20 somatic mutations were unique to samples from invasive cancer (Fig. 1a). We also identified an average of five shared copy number alterations between noninvasive and invasive components, as well as an average of one copy number alteration unique to samples from invasive cancer. A similar mean proportion of somatic mutations and copy number alterations were unique to invasive samples (0.28 for somatic mutations, 0.34 for copy number alterations).

Analysis of our combined whole exome and targeted sequencing data provided multiple insights into IPMN and MCN tumorigenesis. In every analyzed case, there were multiple shared mutations between the noninvasive and invasive components. These included both driver and passenger mutations, indicating that they shared a common phylogenetic ancestor (Fig. 1a, b). In addition, accumulation of unique mutations in both noninvasive and invasive components demonstrated independent evolution after the divergence of the subclone that gave rise to the invasive cancer (Fig. 3, Supplementary Figs. S1–S18). Analysis of additional adjacent low-grade or high-grade samples from the same lesions revealed a subset of shared mutations, suggesting that these dysplastic lesions preceded the development of the invasive carcinoma (Fig. 3, Supplementary Figs. S1, S2, S3, S5, and S16). Evolutionary analyses showed a branched phylogeny in

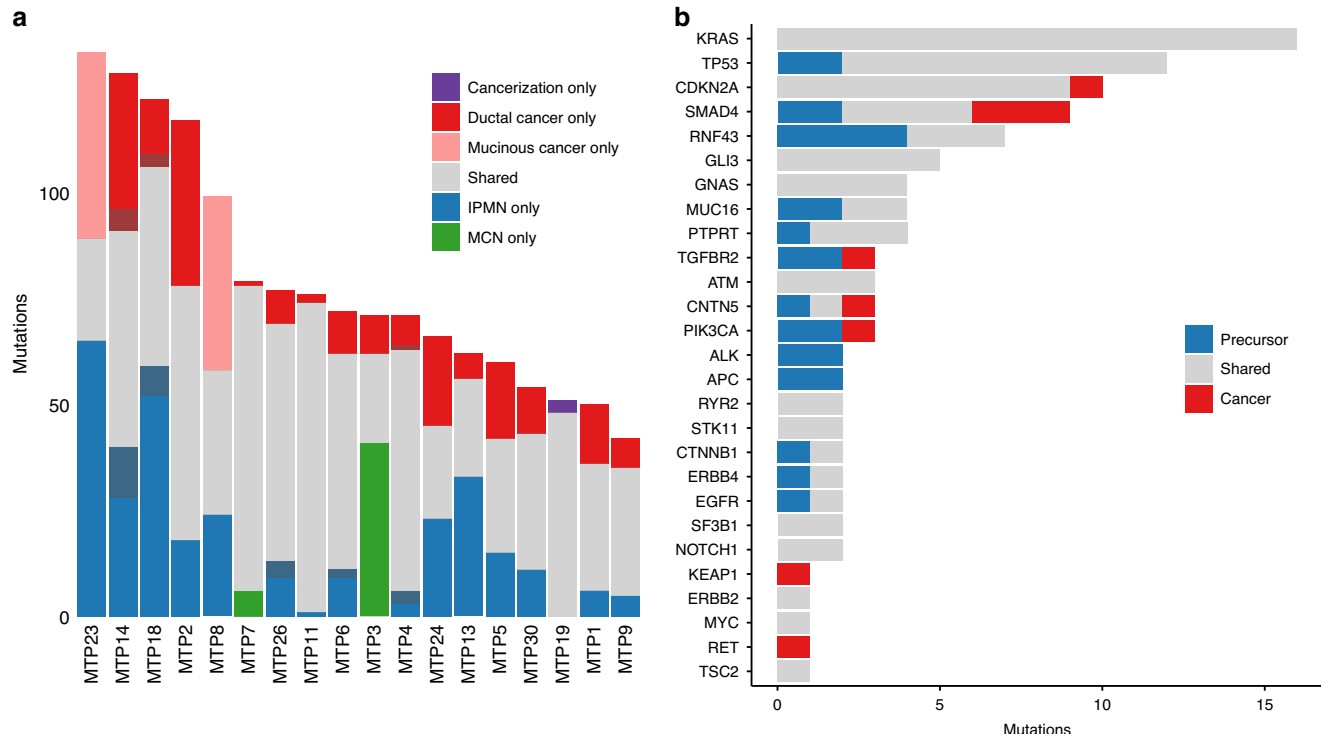

**Fig. 1 Somatic mutations identified in matched noninvasive and invasive cancer samples. a** In each patient sample, multiple mutations were shared between the noninvasive and invasive cancer samples (gray). In addition, some mutations were limited to the noninvasive (blue/green), while others were limited to the cancer (red/pink). Darker colors indicate alterations that were likely restricted to one component but where sequencing coverage in the second component was limited. The proportions of shared and distinct mutations varied between different lesions. **b** Somatic mutations in the most frequently mutated genes are categorized as shared between noninvasive and cancer (gray), limited to noninvasive (blue), or limited to cancer (red). Mutations in some genes (such as *KRAS*) were always shared, while others were enriched in samples from noninvasive (*RNF43*) or cancer (*SMAD4*).

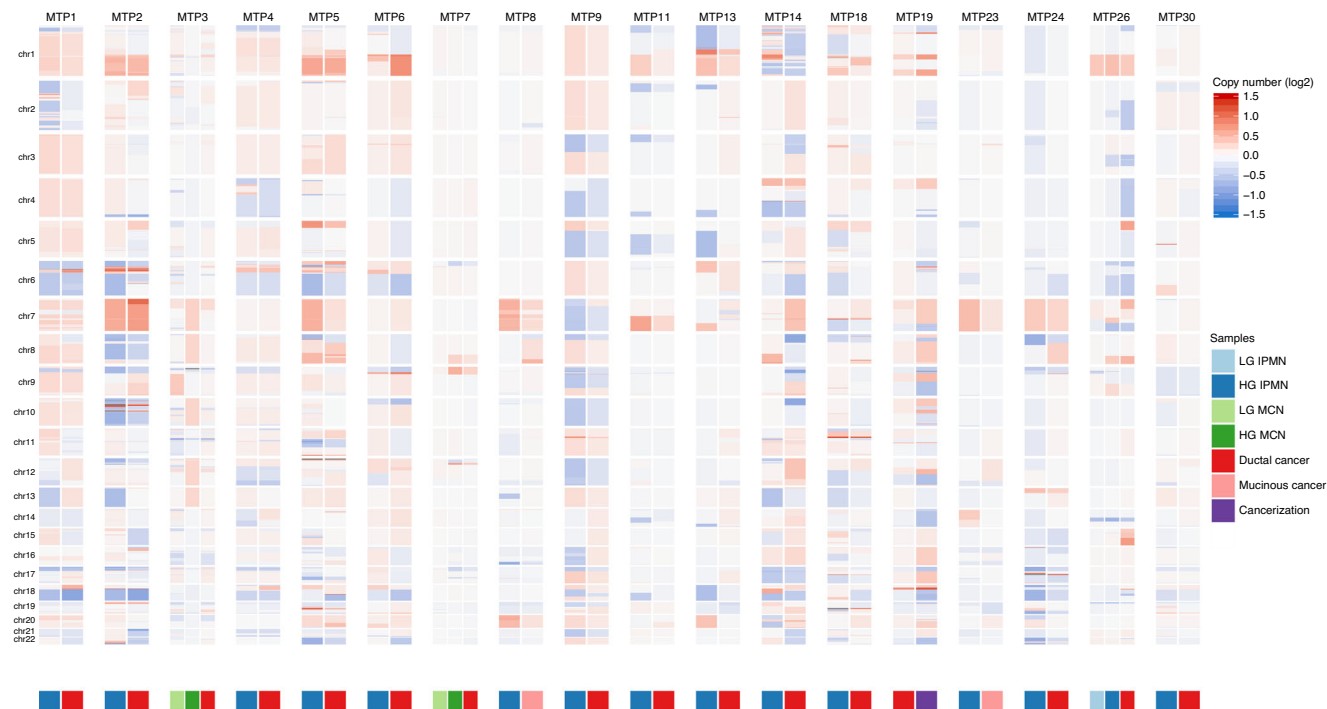

**Fig. 2 Copy number alterations identified in matched noninvasive and invasive cancer samples.** Chromosomal gains (red) and losses (blue) are shown for each chromosome in each patient, with noninvasive samples on the left and cancer samples on the right.

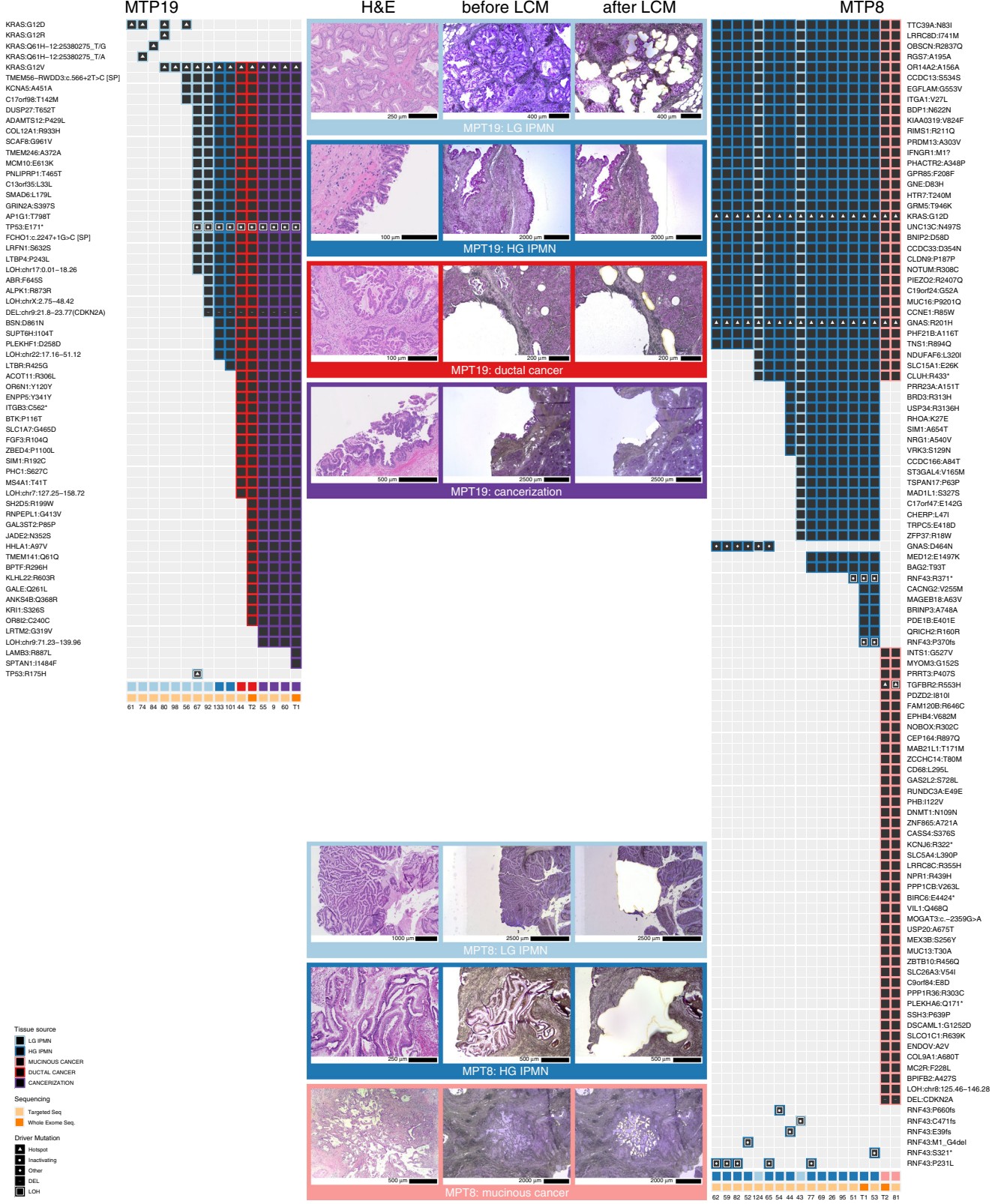

**Fig. 3 Somatic mutations identified in MTP19 and MTP8 in targeted and whole exome sequencing.** We show mutations identified in each sample, including low-grade IPMN (light blue), high-grade IPMN (dark blue), ductal cancer (red), mucinous cancer (pink), and cancerization (purple). The type of sequencing analysis (targeted or whole exome) performed for each sample is indicated in a track on the bottom. Representative images of neoplastic tissue stained by hematoxylin and eosin, as well as isolated regions before and after laser capture microdissection are shown for MTP19 and MTP8.

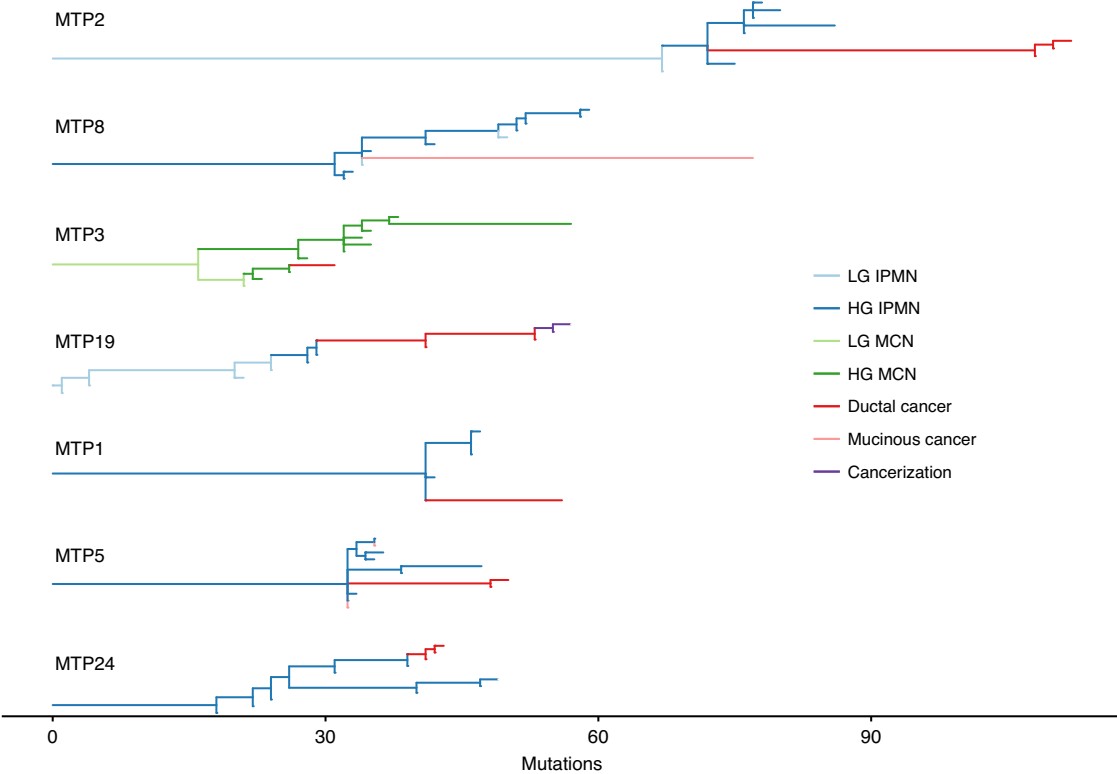

**Fig. 4 Evolutionary reconstruction of samples analyzed by whole exome and targeted sequencing.** In all cases, noninvasive samples (blue/green) precede invasive samples (red/pink) in the evolutionary history. In MTP5, different invasive cancer samples are placed in different regions of the phylogeny, highlighting multiple independent invasion events in this lesion. In MTP19, a sample of cancerization (purple) has descended from invasive cancer samples.

each case, with multiple clonally related but distinct dysplastic samples from each neoplasm (Fig. 4). Importantly, removal of driver gene mutations from these analyses did not significantly alter the resulting phylogenies, indicating that the evolutionary relationships are supported by mutations in addition to driver alterations (which might be shared by chance). Thus, the inferred evolutionary relationships between IPMN/MCNs and cancer samples are robust, as the probability of sharing a non-hotspot mutation due to chance alone is vanishingly small (1 in ~50 million), while the mean number of shared point mutations was 47 (the vast majority of which did not occur in hotspots).

Another potential explanation for shared mutations in noninvasive and invasive samples is the presence of a small number of cancer cells (contamination) in IPMN/MCN samples. Our detailed pathological characterization and macro- or micro-dissection minimized the risk of this sample impurity. In addition, variant allele frequencies (VAFs) of shared mutations in IPMN/MCN and cancer samples can also help to evaluate the likelihood of such contamination. In mutations shared between the IPMN/MCN and cancer, the mean VAF in the noninvasive samples was 0.38, with only 4% of shared mutations having a VAF below 0.1 in the noninvasive samples. These high VAFs indicate that the shared mutations were not the result of a small number of cancer cells contaminating the noninvasive samples. Overall, these data provide evolutionary evidence that IPMNs and MCNs were precursors to invasive pancreatic cancer, with low-grade regions usually preceding high-grade regions and ultimately resulting in invasive carcinoma.

**Driver genes of IPMN/MCN tumorigenesis**. Through whole exome and targeted sequencing analyses of 18 IPMNs/MCNs and associated invasive carcinomas, we confirmed the high prevalence

of mutations in previously identified pancreatic driver genes, including mutations of *KRAS* (89% of cases), *GNAS* (28%), *CDKN2A* (44%), *TP53* (67%), *SMAD4* (50%), and *TGFBR2* (17%) (Fig. 1b). Somatic mutations were also identified in *RNF43* (56%), which has been previously highlighted for its role as a driver in mucin-producing pancreatic cysts[4]. Somatic mutations were observed at low prevalence in key positions in the PI3K (*PIK3CA*, *TSC2*) and WNT (*APC*, *CTNNA2*, *CTNNB1*) signaling pathways as well as in *STK11* (Fig. 1b, Supplementary Data 4). Alteration of these genes and pathways has been previously reported in a fraction of IPMNs[4,7,13,14]. The two MCNs analyzed were similar to the IPMNs in the cohort, with hotspot mutations in *KRAS*, homozygous deletion of *CDKN2A*, and inactivating mutations in *RNF43*, among others, but as expected these MCNs did not have *GNAS* alterations (Supplementary Figs. S3, S7)[6].

In addition to driver genes previously reported in IPMNs, our data provide an opportunity to discover novel drivers of IPMN tumorigenesis. We identified somatic mutations in the DNA damage response gene *ATM* in 17% of lesions, including one nonsense mutation (Fig. 1b, Supplementary Data 4). In addition we identified alterations in Hedgehog pathway member *GLI3* in 5 of 18 cases (28%) (Fig. 1b, Supplementary Data 4). We also identified somatic mutations in a previously described hotspot in *SF3B1*, which encodes a protein critical for RNA splicing (Fig. 1b, Supplementary Data 4). Amplifications of the well-characterized driver genes *ERBB2* and *MYC* were each observed in a single case and have not been previously reported in IPMNs (Supplementary Data 5). Other altered genes with a previously unknown role in IPMN tumorigenesis include *MUC16* (four cases), *PTPRT* (four cases), and *CNTN5* (three cases) (Fig. 1b). Intriguingly, in one case, an *STK11* mutation was found in combination with biallelic *ATM* loss and cancer-specific biallelic *KEAP1* loss—the

combination of these three mutations has previously been reported in lung cancers (Supplementary Fig. S3)[15].

Although *KRAS* mutations occur in the majority of IPMNs and are thought to initiate tumorigenesis in these lesions, two IPMNs lacked mutations in this gene. One case contained a hotspot mutation in codon 227 of *GNAS*, another potential initiator of IPMN tumorigenesis, as well as alterations in *TP53* and *RNF43* (Supplementary Fig. S15). The other case lacked mutations in any of the frequently altered pancreatic driver genes but contained hotspot mutations in both *CTNNB1* (S45P) and *SF3B1* (H662Q) (Supplementary Fig. S10). These cases highlight alternative pathways of initiation and progression in IPMNs lacking *KRAS* mutations.

**Order of genetic alterations in IPMN/MCN tumorigenesis**. Our multiregion sequencing approach of IPMNs/MCNs and associated invasive carcinomas provided insights into the order of specific genetic alterations in pancreatic tumorigenesis. In 17 of the 18 cases, at least one somatic mutation in the initiating driver genes *KRAS* and *GNAS* was shared between the noninvasive component and associated invasive cancer, with the remaining case lacking mutations in these genes. Somatic mutations in *TP53* and *CDKN2A* were also shared in the noninvasive component and associated invasive cancer in the majority of cases. In contrast, *SMAD4* had alterations confined to the invasive carcinoma in three cases and was shared between noninvasive and invasive samples in four cases (Fig. 1b). The majority of *SMAD4* alterations in all sample types were bi-allelic, including 5/7 in noninvasive samples and 6/7 in invasive samples (Supplementary Data 7). Alteration of *TGFBR2*, which functions in the same signaling pathway as *SMAD4*, was also restricted to the cancer in one case (Fig. 1b). The other genes with mutations restricted to the invasive cancer (*CDKN2A, CNTN5, PIK3CA, KEAP1*, and *RET*) only had this pattern in a single sample (Fig. 1b).

Our study also identified driver mutations in subclones of noninvasive neoplasms that diverged from and were not present in invasive cancer. These included hotspots mutations in well-characterized oncogenes and inactivating mutations in tumor suppressor genes (e.g., *PIK3CA* p.E545K, *CTNNB1* p.S45F, *SMAD4* p.E33fs, and multiple inactivating *RNF43* mutations in patient MTP3) (Supplementary Fig. S3). Mutations in *RNF43* were a particularly striking finding in these cases, as some noninvasive components contained several different *RNF43* mutations, each limited to a small number of sections and none involving the invasive cancer (Fig. 3, Supplementary Fig. S3). In addition to heterogeneity in *RNF43* in early lesions, we also identified two cases with multiple mutations in *KRAS* in precursor lesions, of which only one was present in the invasive cancer. For example, in MTP19, *KRAS* p.G12V was present in the majority of IPMN samples, as well as all the invasive cancer samples, but there were an additional four other *KRAS* mutations (all occurring in hotspot positions) that were present in a small number of sections in low-grade IPMN samples (Fig. 3). Intriguingly, these three low-grade IPMN regions shared no mutations with the invasive cancer, suggesting that they represented genetically independent clones.

Notably, while there were often many differences in the somatic point mutations identified in the matched noninvasive and invasive samples, the copy number profiles were quite similar between IPMN/MCNs and invasive cancers (Fig. 2, Supplementary Data 6), and the proportion of copy number alterations unique to cancer samples (0.34) was similar to that observed for somatic mutations. While homozygous deletion of some genes occurred in the invasive cancer but not the noninvasive component, such as *CDKN2A* in MTP8 (Fig. 3), analyses of

chromosomal gains and losses through assessment of allelic imbalance revealed that an average of 91% of the genome was similar in copy number in matched noninvasive and invasive samples (Fig. 2, Supplementary Data 6).

**Insights into pancreatic neoplasia revealed by sequencing**. The samples analyzed by targeted sequencing were characterized morphologically and meticulously isolated using laser capture microdissection. Even with this process, we identified samples in two cases that were characterized morphologically as IPMNs but through genomic and evolutionary analyses were determined to be identical to or descendants of the associated invasive cancers. For example, in MTP19, some of the samples originally identified morphologically as noninvasive IPMN (55, 9, 60, and T1) shared all the mutations present in the invasive cancer sample (T2) and contained additional mutations, suggesting that these samples descended from the cancer (Figs. 3, 4). Evolutionary analyses suggested that some of the morphologically identified IPMN samples in this case (as well as MTP1) actually represented intraductal spread of invasive carcinoma, also referred to as cancerization of the ducts. In these cases, after invading the stroma, the carcinoma invaded back into and colonized the cyst such that it was morphologically indistinguishable from IPMN with high-grade dysplasia.

In one case (MTP5), we also identified an interesting pattern of multifocal invasion of the carcinoma. In this case, we analyzed five different samples from invasive cancer—three samples were isolated from a mucinous carcinoma, and two samples were isolated from a ductal carcinoma. Based on evolutionary analyses of the patterns of shared and distinct mutations in the cancers and IPMNs, we conclude that there were multiple separate invasion events in this lesion, as represented by the mutations shared between the invasive cancers and noninvasive components as well as those that were unique to the specific invasive cancers (Fig. 4, Supplementary Fig. S5).

As our study represents the largest cohort of comprehensively sequenced IPMNs/MCNs, we also analyzed mutational signatures in our dataset. Intriguingly, our data contrast somewhat with the mutational signatures previously reported in pancreatic ductal adenocarcinoma (PDAC)[16,17]. Like PDAC, the most prominent mutational signature was associated with age (Signature 1A), which was identified in almost every case (Supplementary Fig. S19). However, we also identified signatures associated with APOBEC enzymes (four cases), smoking (three cases), and mismatch repair deficiency (11 cases). Although smoking is considered a risk factor for pancreatic cancer, until now the mutational signature associated with smoking has not been reported in pancreatic neoplasia[18].

**Evolutionary timeline of high-grade IPMN to PDAC**. To estimate the time between the development of high-grade IPMN and PDAC, we evaluated Bayesian hierarchical models for the number of acquired mutations under a range of possible mutation rates. These models estimate the time interval between a founder cell of a PDAC and the ancestral precursor cell in the associated high-grade IPMN assuming that mutation rates and cell division times are constant throughout this period of development (see "Methods"). We performed this analysis on the paired WES data from 17 of our 18 cases (Supplementary Fig. S20). We excluded MTP19 because our evolutionary analyses demonstrated intraductal spread of invasive carcinoma and as such, we lacked WES data from an IPMN sample in this case. In the 17 analyzed cases, the average median time to progression from IPMN to PDAC was 3.7 years, but the models showed a bimodal distribution. This median time was nearly 3 years for 13 patients, but nearly 7 years

for 4 patients with more than 35 acquired mutations, highlighting potential variability in progression time between patients. For example, in patient MTP1, most models suggested an average of 2.8 years between the development of the IPMN and the PDAC (90% CI, 1.3–6.7 years). In contrast, for patient MTP2 with 36 additional mutations acquired in the PDAC, the transition appears to have been slower with an average estimate of 6.6 years (90% CI: 3.9–11.4 years) from the Bayesian models. Overall, these analyses suggest that for most patients there is a significant window of time between development of high-grade dysplasia and pancreatic cancer, providing an opportunity for surveillance and intervention.

## Discussion

This study represents the largest dataset of whole exome sequencing of IPMNs and MCNs to date. Importantly, our data established that both IPMNs and MCNs are direct precursors of invasive pancreatic cancer (Fig. 4). This conclusion has been previously suggested by the morphological relationship between the noninvasive neoplasms and invasive cancer on traditional histologic sections, as well as shared driver gene mutations in targeted sequencing studies[6,7,9–11]. However, the presence of many shared driver and passenger mutations clearly demonstrated the common origin of IPMNs/MCNs and invasive pancreatic cancers in our study, and evolutionary analyses revealed that dysplastic lesions precede invasive cancers. Evolutionary analyses suggested that high-grade noninvasive lesions occur over 3 years before invasive carcinoma, providing a window of opportunity for early detection and intervention.

In this study, we identified somatic mutations in driver genes that had not been previously implicated in IPMNs/MCNs. For example, we identified alterations in the DNA damage response gene ATM in 17% of the analyzed cases. Germline mutations in ATM have been recently reported in patients that developed IPMNs, highlighting the potential importance of this gene in IPMN risk[19]. In addition, mucinous (colloid) carcinomas are significantly more common than typical ductal carcinomas in patients with germline ATM mutations, further highlighting the link between mutations in this gene and IPMNs[20]. Although the ATM gene is large, potentially increasing the likelihood of passenger or artifactual mutations, even larger genes such as TTN had a lower mutation prevalence, suggesting that at least some of the alterations identified in ATM are likely to be bona fide somatic mutations. Somatic mutations in GLI3, which encodes a component of the Hedgehog signaling pathway, were identified in 28% of cases. Somatic mutations in GLI3 were recently reported in a distinct morphological variant of pancreatic carcinoma (undifferentiated carcinoma with osteoclast-like giant cells) as well as at a low prevalence in sporadic PDAC, suggesting that the importance of GLI3 mutations and its signaling pathway in pancreatic tumorigenesis may extend beyond IPMNs/MCNs[21–23]. The hotspot mutations in SF3B1, which encodes a protein critical for RNA splicing, are also potential drivers in the IPMN pathway. However, somatic mutations in this gene have been reported in a variety of other neoplasms, including hematologic malignancies and uveal melanoma[24–26].

We highlight somatic alteration of the SMAD4 pathway as a putative driver of progression to invasive cancer, as mutations in SMAD4 or TGFBR2 occurred only in invasive cancer samples in 4 of the 18 cases analyzed. SMAD4 was the only gene with cancer-specific mutations in more than one case, highlighting the potentially unique role this gene plays in pancreatic carcinogenesis. This role has been previously suggested by next generation sequencing of high-grade PanINs showing an absence of SMAD4 mutations in precancerous lesions, as well as cancer-specific

SMAD4 mutations reported in a paired PanIN/carcinoma analyses[27,28]. Loss of SMAD4 expression limited to invasive carcinomas has been reported in MCN- and IPMN-associated invasive cancers, and targeted sequencing of a small number of IPMNs and matched cancers identified a single case with a SMAD4 mutation occurring only in the cancer[7,28,29]. In our data, there were also four cases where mutations in SMAD4 were shared between noninvasive and invasive cancer samples, and two where SMAD4 mutations were limited to the noninvasive component. Although our whole-exome approach could not detect all types of SMAD4 alterations (such as rearrangements or epigenetic changes), the majority of SMAD4 mutations observed in both noninvasive and invasive components affected both alleles. Taken together, this suggests that the role of SMAD4 mutations may not be universal and may depend on other factors, including cell intrinsic (such as somatic mutations in other driver genes) and cell extrinsic (such as stromal and immune microenvironment) mechanisms.

Although some of our cases had SMAD4 mutations limited to the invasive cancer, most of the IPMN/MCN-associated cancers lacked driver gene alterations that were associated with invasive disease, suggesting that malignant progression is not universally driven by point mutations. Previous studies have specifically demonstrated the importance of copy number alterations and chromosomal rearrangements in pancreatic tumorigenesis[30]. We did not identify large differences in the copy number profiles between noninvasive components and associated invasive cancers, suggesting that global genomic instability may be important as an early feature of tumorigenesis but is not likely to drive malignant transformation in many cases.

Our study also revealed prevalent genetic heterogeneity in driver gene mutations in early lesions, demonstrating more complex processes than previously suggested by traditional linear tumorigenesis models. Similar to our recently reported polyclonal origin of IPMNs[12], we identified multiple independent clones initiated by distinct KRAS mutations in two cases in the current study. In addition, our study identified multiple distinct inactivating mutations in RNF43 limited to unique tumor subclones, a pattern previously observed by our group and not shared by other genes in our whole exome sequencing analyses[12,31]. In most cases, RNF43 mutations were enriched in noninvasive components and absent from the associated invasive cancers. More generally, we observed multiple instances of clear driver mutations (including hotspot mutations in oncogenes as well as inactivating mutations in tumor suppressor genes) that were limited to the noninvasive components and not present in the associated invasive cancer. Thus, these mutations occur and clonally expand in the IPMN or MCN but are not present in the subclone that subsequently invades. Such independent evolution of premalignant lesions has been observed in other organ sites and does not diminish the conclusion of our evolutionary analyses that IPMNs/MCNs are precursors of invasive pancreatic cancer[32–34]. Rather, these observations suggest unique selective processes at different time points in tumorigenesis, such that mutations selected in the precancerous lesion are not selected for (or are even selected against) in the invasive cancer.

In addition to these observations about clonal evolution in noninvasive lesions, our data also provide genetic evidence for multiple underappreciated processes in pancreatic neoplasia. First, we provide genetic evidence for intraductal spread of invasive carcinoma, also known as cancerization[35]. In two of our cases (MTP1 and MTP19), the identified somatic mutations suggested that samples that were morphologically thought to be IPMN were actually of the same clone or clone descended from invasive cancer. These cases confirmed the morphological impression of the prevalence of this cancerization phenomenon,

which likely has confounded many previous studies of precancerous pancreatic lesions[35]. In addition, we describe one case of IPMN with multiple independent invasion events (MTP5). This case contained invasive cancer with two different morphologies, one with typical ductal morphology and one with mucinous (colloid) morphology. Evolutionary analyses demonstrated that the ductal and mucinous carcinomas arose through independent invasion events and suggested that the multiple mucinous cancer samples comprised unique subclones that invaded independently from the IPMN. Although multifocal invasion has been described morphologically in IPMNs with multiple anatomically discrete invasive foci, in this case all the invasive carcinoma samples came from the same grossly defined tumor, suggesting that multifocal invasion may be an under-appreciated phenomenon in IPMNs.

The results of our study should also be considered in the context of studies of other precancerous lesions and associated invasive cancers. Previous molecular studies of IPMNs and associated invasive carcinomas have been limited to targeted gene panels[9–11]. These studies demonstrated a small number of shared and distinct driver gene mutations in lesions from the same patient. The results of these studies are largely consistent with our study, but the limited panels prevent comprehensive evolutionary conclusions. In addition, complementary studies that have employed whole exome sequencing to characterize microscopic pancreatic precancerous lesions (PanINs) and their co-occurring invasive carcinomas highlight the common evolutionary origin of PanINs and co-occurring PDACs[27,36]. Similar to our study, these studies reported a lack of consistent specific driver genes associated with invasive cancer, although cancer-specific SMAD4 mutations were reported in two cases in one study[27]. Importantly, in our study the sequencing of additional precancerous samples (beyond the original paired samples analyzed by whole exome sequencing) provided a more detailed analysis of precancerous clonal evolution than previous studies.

As with all genomic analyses, our study does have some limitations. Compared to other genomic analyses of invasive pancreatic cancer, the sample size in our study is relatively small, with multi-region whole exome sequencing performed on 18 patients. Nevertheless, the whole exome and targeted analyses of 148 samples from these patients represents the largest genomic study of precancerous pancreatic lesions to date. In addition, our combined approach of whole exome and targeted sequencing may not have identified all mutations in all regions of these comprehensively analyzed IPMNs. Despite these limitations, the analyses provide a detailed view of the acquisition of mutations that characterize the invasive carcinoma, as well as genetic heterogeneity in well-characterized pancreatic driver genes. Finally, in this study, we focused entirely on genetic alterations, as the role of these mutations in driving tumorigenesis has been well documented. Our study provides evidence that the evolution to invasive cancer is likely to be driven by non-genetic mechanisms in some lesions, highlighting an important direction of future investigation.

In this study, we present a comprehensive evolutionary analysis of precancerous pancreatic cysts and associated invasive carcinomas. We demonstrate that IPMNs and MCNs are precursors of invasive pancreatic cancer, that alterations in ATM, GLI3, and SF3B1 are present in these lesions, and that SMAD4/TGFBR2 alterations are likely drivers of invasion in a subset of cases. Analyses of the evolutionary timeline between high grade precancerous lesions and pancreatic cancer suggest a window of more than 3 years for acquisition of these invasive characteristics. These data provide critical insights into pancreatic tumorigenesis and highlight an opportunity for surveillance of precancerous pancreatic cysts and early detection of pancreatic cancer.

## Methods

**Specimen acquisition.** An IRB protocol was submitted and approved at the Johns Hopkins University. Additional IRB protocols were submitted and approved in collaborating institutions when a local IRB was required. Samples came from the Johns Hopkins Hospital, Baltimore (United States); University Medical Center, Utrecht (The Netherlands); University and Hospital Trust, Verona (Italy); Asan Medical Center, Seoul (Republic of Korea); National Cancer Center Hospital, Tokyo (Japan); Royal North Shore Hospital, Sydney (Australia), University Hospital, Ghent (Belgium); Academic Medical Center, Amsterdam (The Netherlands); Laboratory for Pathology Eastern Netherlands, Hengelo (The Netherlands); Thomas Jefferson University, Philadelphia (United States); Aichi Cancer Center Hospital, Nagoya (Japan); Medica Sur Clinic and Foundation, Mexico City (Mexico); Emory University Hospital, Atlanta (United States). Informed consent was obtained from all participants in accordance with IRB requirements. The pathology archives were searched for cases meeting the following inclusion criteria: synchronous IPMN or MCN and pancreatic cancer, whereby the IPMN or MCN is larger than the invasive component and the invasive component is smaller than 2.5 cm. We specifically sought small pancreatic cancers to isolate the alterations that occur at the time of malignant progression (rather than those that accumulate during further growth of the invasive cancer). The H&E slides were evaluated by experienced pathologists with a subspecialty in gastrointestinal pathology (L.D.W., R.H.H., L.A.A.B., G.J.O.), and the different components (noninvasive cyst and invasive cancer) were annotated.

**Sample preparation and whole exome sequencing.** The formalin-fixed paraffin-embedded (FFPE) tissue blocks were cored separately for the different components: high-grade IPMN/MCN, invasive cancer, and matched normal sample from the duodenum or spleen. As these are human tissue blocks from pathology archives of the participating hospitals, these materials are not available to the public. In three cases, a low-grade IPMN/MCN was also cored. DNA was extracted from these cases using the Qiagen QIAamp DNA FFPE Tissue Kit according the manufacturer's protocol (Qiagen). After purification, the final DNA concentration was measured with the Qubit 2.0, dsDNA high sensitivity assay. Noninvasive lesions invariably had a higher neoplastic cellularity, as assessed during pathology review. Twenty cases were selected for whole-exome sequencing (WES) of the noninvasive component(s), cancer, and matched normal at Personal Genomic Diagnostics (PGDx, Baltimore). Briefly, genomic DNA was fragmented, followed by end-repair, A-tailing, adapter ligation, and polymerase chain reaction. PCR products were purified, and exonic regions were captured in solution using the Agilent Sure Select kit according to the manufacturer's instructions (Agilent). Pair-end sequencing was performed on a Hi-Seq2500 next-generation sequencing instrument (Illumina). Primary processing of sequence data was performed using Illumina CASAVA software (v1.8). Two cases (MTP15 and MTP20) failed quality control during sequencing and were excluded from further analysis. Fastq-files were further processed according to the GATK best practices workflow: BWA (v0.7.17) for alignment to hg19, picard (v2.18.1) for duplicate reads flagging and GATK (v3.7) for base quality score recalibration.

**Somatic mutation identification from whole exome sequencing data.** Non-synonymous mutations were called with PGDx's VariantDX mutation caller, and synonymous mutation were called using Mutect2 (in GATK v3.7), both using default parameters[37,38]. All identified mutations were confirmed by visual inspection using the Integrated Genomics Viewer (v2.3.80; IGV)[39]. In order to assess mutated sites in additional samples from the same patient, a separate analysis was performed with Manta (v1.4.0) and Strelka (v2.9.9)[40]. To obtain information on the mutated positions across all samples from each patient, Strelka was used to obtain metrics at each mutated position in all matched samples. Then, mutations were filtered based on the several metrics. A mutation was called when the normal sample had a depth of at least 10 distinct reads and the mutation found had a mutant allele fraction of <2% in the normal. Further filtering is described in "Integrated mutation analysis" below. In addition, analysis of each candidate mutation was performed using BLAT (v36)[41]. A sequence of 101 bases (with the mutated position in the middle) was used as the query sequence (http://genome.ucsc.edu/cgi- bin/hgBlat). Candidate mutations were removed from further analysis if the analyzed region resulted in >1 BLAT hits with 90% identity over 70 SCORE sequence length. All candidate alterations were verified by visual inspection in IGV.

In order to analyze the mutational signatures, we combined the single nucleotide variants found in both the noninvasive sample(s) and the invasive sample into a set of unique mutations. Each mutation was classified into one of the 96 trinucleotide contexts[18]. The contribution of each signature to each tumor sample was estimated using the deconstructSigs (v1.8.0) R package.

**Sample preparation and targeted sequencing.** Additional blocks were available from seven of the analyzed IPMNs/MCNs, allowing multi-region sampling of both the precursor lesion and the pancreatic cancer. Twenty to thirty serial sections of FFPE tissue from each available block were cut onto membrane slides (PEN 1.0, Zeiss, Oberkochen, Germany) for laser capture microdissection. After deparaffinization in xylene and rehydration in ethanol, the slides were stained with hematoxylin.

Morphologically homogeneous regions were microdissected using the LMD7000 (Leica, Wetzlar, Germany). Genomic DNA was extracted from the microdissected tissues. In brief, the microdissected samples were incubated with proteinase K for 16 h at 56 °C. The digested mixture was transferred to a 130 μL microtube (Covaris, Woburn, MA) for shearing. Following fragmentation, the sample was further digested for 24 h at 56 °C followed by 1 h incubation at 80 °C to inactivate proteinase K. DNA purification was then performed using the QIAamp DNA FFPE Tissue kit following the manufacturer's instructions (Qiagen, Valencia, CA). Fragmented genomic DNA from tumor and normal samples were used for library preparation according to the Agilent Sure Select Target Enrichment System (Agilent, Santa Clara, CA). The targeted panel (Agilent) was constructed, containing all the regions with the mutations identified in the WES from these seven cases. The protein coding regions of 15 pancreatic cancer driver genes (*APC, ATM, BRAF, CDKN2A, CTNNB1, GNAS, KRAS, MAP2K4, PIK3CA, PTEN, RNF43, SMAD4, STK11, TGFBR2, TP53*) were also included in the targeted panel[12]. Library preparation was performed using the Agilent Sure SelectXT Target Enrichment System (Agilent) following the manufacturer's instructions with the following modifications: genomic DNA was sheared prior to library preparation (see above). In addition, half of the volume of capture library was used per reaction, and supplemented with water. Finally, both pre- and post-capture PCR cycle number was increased by 0 or 1 for DNA inputs of 200 ng or 50–100 ng, respectively. Paired-end sequencing was performed using the Illumina MiSeq (Illumina, San Diego, CA).

**Somatic mutation identification from targeted sequencing data.** Samples with an average sequencing depth of >150× in the targeted regions were included for further analysis. Mutations were identified by Strelka, specifying the union of mutations found in the samples of the each patient as input for a second round of analysis[40]. This approach yielded mutation metrics (reference and alternate read counts) across all samples from each patient. Only mutations identified in the whole exome data of each patient or mutations found in the 15 driver genes that were fully sequenced were considered for further analysis. Newly discovered mutations in the fully sequenced genes were visually inspected in IGV.

**Integrated mutation analysis.** The following criteria were applied to establish presence or absence of mutations in each analyzed sample profiled by whole exome or targeted sequencing, which subsequently informed the phylogenetic analysis. To start, the median depth of coverage over the capture region was calculated for each sample. The mutated loci selected for analysis in each patient were required to have a sequencing depth of coverage exceeding 20% of the median coverage of the sample, in all sequenced samples of the patient. This strategy enabled exclusion of loci where insufficient or inconsistent capture efficiency might restrict the power to detect mutations. In a whole exome sequenced samples, a minimum mutant allele fraction of 10% and five mutant reads was used to define mutation presence (primary mutation calls). However, our analysis of tumor purity revealed a marked difference between the level of normal contamination in precursor and cancer lesions. Given that reduced tumor purity results in lower levels of mutant allele fraction and mutant read counts, for mutations marked as present in at least one sample of each patient, a minimum of two mutant read counts was deemed sufficient to call the mutation present in a whole exome sequenced samples (secondary mutation calls). In samples analyzed by targeted sequencing, a minimum mutant allele fraction of 5% and a sample-specific minimum mutant read count were applied to mark a mutation as present. The lowered mutant allele fraction was applied to take advantage of the fact that the higher depth of coverage in targeted sequencing can allow sensitive mutation detection at lower MAF levels without compromising accuracy. The sample-specific minimum mutant read count was obtained by scaling the value 5 (threshold used for whole exome sequenced sample), by the ratio of the median coverage of the targeted sample to the average median coverage of the whole exome samples of that patient. Increasing the minimum required value for the mutant read count helped avoid false positive calls due to sequencing errors at increasingly high levels of coverage.

**Copy number evaluation.** In samples analyzed by whole exome sequencing, we applied FACETS (v0.5.14) to determine the tumor purity and ploidy of the sample, as well as allele-specific copy number for regions across the genome[42]. In each tumor sample, we investigated focal copy number aberrations (focal CNAs) by focusing on genomic regions with length <3 Mb where the estimated copy number was at zero (homozygous deletion), or was greater than or equal to three times the estimated ploidy of the sample (focal amplifications). We filtered the focal copy number aberrations to those that passed the visual review in the following categories: (1) In samples with tumor purity of 30% or above, any aberrations affecting the 15 driver genes frequently mutated in pancreatic cancer regardless of their estimated cancer cell fraction were reviewed. In addition, we reviewed other focal CNAs with cancer cell fraction of 75% or above (likely clonal). (2) In samples with tumor purity below 30%, we reviewed all changes affecting an in-house set of 195 cancer driver genes if their estimated cancer cell fraction was 75% or above. Finally, we visually inspected each detected focal CNA in the other whole exome sequenced samples from the same individual to enable recovery of potentially false negative calls due to technical issues. This step allowed us to use focal CNAs as an additional class of features in phylogenetic analysis.

In whole exome sequenced samples, the allele specific copy number values estimated by FACETs were used to determine whether a locus harboring a somatic mutation has undergone LOH, as indicated by minor copy number of zero. In seven cases where a number of lesions were profiled by targeted sequencing, the small size of the panel (~185 kb) prohibited analysis by conventional tools. Therefore, we applied a custom analysis pipeline to evaluate the copy number status of the 15 driver genes whose coding sequence was fully covered by the panel as follows. First, the base level depth of coverage for all positions on the panel were calculated and summarized by taking the median across each interval on the panel. Next, these values were then normalized by the median of the coverage for all the intervals on the panel, and corrected for GC content. The resulting copy ratio values were aggregated for each driver gene by taking the median of all corresponding genomic intervals. Finally, homozygous deletion was defined as a copy ratio value below −1.25 in log2 scale (using a 0.25 margin of error to allow differentiation from hemizygous deletion), and focal amplification was defined as a copy ratio value exceeding 1.6 in log2 scale (corresponding to copy number 6 or above in a largely diploid genome).

**Genome-wide assessment of allelic imbalance and LOH.** A comparative analysis of LOH across the tumor samples of each patient was performed to identify structural alterations occurring in the course of tumor evolution as described previously[32]. Given the difference in the breadth of genome coverage in whole exome and targeted sequenced samples, the number of informative loci (germline heterozygous SNPs) widely varies between the two approaches. Therefore, our analysis started by evaluation of LOH in whole exome samples and was later extended to targeted samples in regions were SNPs with sufficient coverage were available in targeted sequencing data. Circular binary segmentation was applied to the minor allele frequency (mAF) of germline heterozygous SNPs in each whole exome sequenced sample to determine genomic region with a constant level of allelic imbalance[43]. In each sample, a difference of 0.1 between the segment mAF of tumor and matched normal samples was required to label a region as harboring LOH. Across genomic segments with lengths exceeding 10 Mb and overlapping at least 20 SNPs, the minimum of segmental mAF was recorded and used as proxy measure reflecting the purity of the tumor sample. The confidence in LOH calls for each segment was determined by comparison of the segment mAF with the minimum sample-level mAF and reported in three tiers: (1) high confidence tier: segment mAF is within 0.1 distance of minimum sample-level (2) intermediate confidence tier: segment mAF distance to minimum sample-level maf is in 0.1–0.2 range (3) low confidence tier: segment mAF exceeds the minimum sample-level mAF by at least 0.2. Next, the union of the genomic coordinates for segment breakpoints across all whole exome sequenced samples of each patient was derived and the segmental mAF and the number informative SNPs in each sample was calculated in the intervals defined by each pair of consecutive breakpoints (patient-level segments). The resulting segments were filtered to those spanning a minimum of 10 Mb, overlapping 20 informative SNPs, and belonging to the high confidence LOH tier in at least one sample of the patient. The segments passing the above filters were categorized into two sets: those uniformly present across all, and those differentially present in a subset of whole exome sequenced samples. To increase the specificity, the latter set was further narrowed down to the set where there was a minimum difference of 0.1 between the segmental mAF of samples with and without LOH. To avoid overestimation of the number of independent structural alterations, LOH segments with boundaries within a 5 Mb window and identical LOH calls across all samples analyzed were merged together.

This analysis was extended to include targeted sequenced samples as follows. First, the number of reference and alternate alleles at each germline heterozygous SNP were determined and the SNPs were filtered to those with minimum distinct coverage of 20×. For each segment, the count and average mAF of the overlapping SNPs were calculated. LOH of each candidate segment from whole exome analysis was determined using the criteria described above, and segments with LOH were classified into the three confidence tiers. This approach yielded a candidate set of LOH changes which can be used in conjunction with somatic mutations to inform the phylogenetic analysis. Moreover, the LOH status across the genome can be used to annotate mutations in each sample, and highlight cases where the absence of mutation in a sample is due to structural alterations as opposed to intra-tumoral heterogeneity across the samples.

**Phylogenetic analysis.** To derive a parsimonious description of tumor evolution in each patient, the set of somatic alterations (mutations, LOH changes, focal CNAs) identified was first represented as binary matrix reflecting their presence in each analyzed sample. The unique patterns of presence/absence observed across samples of each patient were determined and used to define alteration clusters. These clusters comprise of the entire set of alterations with a common pattern of presence across the samples. To avoid spurious assignment of singular alterations to individual clusters due to false positive or negative calls, the initial set of clusters were filtered to those with at least two alterations. The remaining set of alterations belonging to singleton clusters were reviewed to determine whether they can be merged with any of the existing clusters, by considering the quality of mutation or LOH calls in samples where they differed; i.e., the distance

# ARTICLE

of mutation/LOH change metrics from the thresholds used to establish the calls and evaluating the possibility of mutation loss due to LOH of the mutated loci and copy number loss. At this point, the presence pattern of each cluster was assumed to represent that of all its constituent mutations and was used to correct any erroneous calls identified through the review process. In 13 patients where all samples were whole exome sequenced and allele-specific copy number was uniformly available across all the samples, mutations where the absence in subset of samples could be explained by either copy number loss or intra-tumoral heterogeneity were excluded from phylogenetic analysis to avoid ambiguity.

Given the focus of the current study on evaluation of heterogeneity and evolution among lesions in each patient and not the clonal heterogeneity within each lesion, the genotype of cancer cells within each lesion was assumed to be uniform and defined by the presence of alterations identified. However, in cases where a subset of samples are defined by co-existence of two or more clonal populations, such an assumption will result in inaccurate phylogenetic reconstruction on the level of samples as the mixed samples cannot be represented by a unique genotype. Therefore, we sought to identify cases with possible clonal mixing and applied a correction method inspired by one of conceptual framework of SCHISM (v1.1.2) as follows[44]. First we evaluated all pairs of mutation clusters and identified the subset where neither of the two members of the pair can be ancestral to the other in the phylogenetic tree; i.e., the set of samples harboring each member of the pair are not nested within each other. Such pairs indicate mutation clusters acquired on distinct branches of the phylogenetic tree, and should not co-occur in any tumor samples in absence of clonal mixing. The set of mutation cluster pairs $(a^i, b^i)$ is narrowed down such that no potential ancestor of $a^i$ is present with $b^i$ as a pair in the set and vice versa. Next, the set is further filtered down to mutation cluster pairs that do co-occur with each other in at least one tumor sample. Finally, any tumor sample harboring both members of such a mutation cluster pair is represented as two distinct populations of cancer cells (subclones) each defined by the presence of one member of the pair and the absence of the other and its potential descendants (mutation clusters present in a subset of samples where the original cluster is present).

The binary matrix resulting from the analysis above were used as inputs to the maximum parsimony phylogeny module (pars) in PHYLIP (v3.695, http://evolution.genetics.washington. edu/phylip/), and the resulting phylogenetic trees were visualized using the ggtree (v1.4.11) module in R.

**Analysis of timing of malignant progression.** Our sampling model for the number of additional mutations acquired in PDAC sample $i$ is Poisson with mean $\theta_i$. The average number of mutations $\theta_i$ can be factored as the product of the mutation rate $\mu$ (assumed to be the same for all patients) and the number of years $T_i$ during which the $y_i$ mutations accumulated. The second stage of the model posits a Gamma sampling distribution for the timing between the birth of the high-grade IPMN and the PDAC. Finally, we use diffuse priors for the shape and rate parameters of the Gamma. The model implemented in JAGS version 4.3.0 is

$$y_i \sim \text{Poisson}(\theta_i)$$
$$\theta_i = \mu T_i$$
$$T_i \sim \text{Gamma}(a, b)$$
$$a = 1 + m * b$$
$$b = \frac{m + \sqrt{m^2 + 4 * s^2}}{2s^2}$$
$$m \sim \text{Uniform}(0, 100)$$
$$s \sim \text{Uniform}(0, 100),$$

where $m$ and $s$ correspond to the mode and standard deviation of the gamma prior for $T$. As the mutation rate $\mu$ is not known, we implemented this model for a range of plausible values (1 mutation/year–10 mutations/year).

**Statistics and reproducibility.** All neoplastic samples were sequenced once: through whole exome sequencing or through targeted sequencing. In cases with both whole exome and targeted sequenced, normal samples were sequenced both with the whole exome probes and the targeted probes. Otherwise, normal samples were sequenced once.

**Reporting summary.** Further information on research design is available in the Nature Research Reporting Summary linked to this article.

## Data availability

When permitted by the relevant IRBs (77 samples), whole exome and targeted sequencing data has been deposited in the European Genome-phenome Archive with accession EGAS00001004473.

## Code availability

Code for the analysis of the timing of malignant progression is available at https://gitlab.com/cancer-genomx/ipmn-timing.

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

## Acknowledgements

The authors acknowledge the following sources of support: NIH/NCI P50 CA62924; NIH/NIDDK K08 DK107781; NIH/NCI R01 CA121113; NIH/NCI R00 CA190889; NIH/NCI P30 CA006973; Sol Goldman Pancreatic Cancer Research Center; Buffone Family Gastrointestinal Cancer Research Fund; Carol S. and Robert M. Long Pancreatic Cancer Research Fund; Kaya Tuncer Career Development Award in Gastrointestinal Cancer Prevention; AGA-Bernard Lee Schwartz Foundation Research Scholar Award in Pancreatic Cancer; Sidney Kimmel Foundation for Cancer Research Kimmel Scholar Award; AACR-Incyte Corporation Career Development Award for Pancreatic Cancer Research; American Cancer Society Research Scholar Grant; Emerson Collective Cancer Research Fund; Rolfe Pancreatic Cancer Foundation; Joseph C Monastra Foundation; The Gerald O Mann Charitable Foundation (Harriet and Allan Wulfstat, Trustees); Susan Wojcicki and Denis Troper; Dr. Miriam and Sheldon G. Adelson Medical Research Foundation; Dutch Digestive Foundation (MLDS CDG 14-02); Nijbakker-Morra Foundation; Lisa Waller Hayes Foundation; Avner Pancreatic Cancer Foundation.

## Author contributions

M.N., L.A.A.B., and L.D.W. designed the study. M.N., W.H., C.L., A.P., A.S., G.B., G.Z., P.C., S.M.H., S.Y., N.H., A.J.G., J.S.S., G.J.A.O., A.H., J.V., C.J., N.V.A., W.J., J.W., J.A.S., B.T., E.D.T., L.A.A.B., and L.D.W. analyzed tissue samples. M.N., C.G.F., W.M.H., W.H., M.D., E.P., V.A., L.A.A.B. acquired data. M.N., N.N., V.B.G., J.R.W., N.J.R., R.K., R.B.S., and L.A.A.B. analyzed data. M.N., N.N., R.H.H., R.K., R.B.S., L.A.A.B., V.E.V., and L.D.W. interpreted data. M.N., V.E.V., and L.D.W. wrote the paper. All authors have approved the submitted version of the paper.

## Competing interests

L.D.W. receives research funding from Applied Materials. V.E.V. is a founder of Personal Genome Diagnostics, a member of its Scientific Advisory Board and Board of Directors, and owns Personal Genome Diagnostics stock, which are subject to certain restrictions under university policy. V.E.V. is an advisor to Takeda Pharmaceuticals. Within the last five years, V.E.V. has been an advisor to Daiichi Sankyo, Janssen Diagnostics, and Ignyta. J.R.W. is founder and owner of Resphera Biosciences LLC, and is a consultant to Personal Genome Diagnostics Inc. The terms of these arrangements are managed by Johns Hopkins University in accordance with its conflict of interest policies. The other authors declare no conflict of interest.

## Additional information

Michaël Noë [1,2], Noushin Niknafs[2], Catherine G. Fischer[1], Wenzel M. Hackeng[1,3], Violeta Beleva Guthrie[4,5], Waki Hosoda[1,6], Marija Debeljak [1], Eniko Papp[2], Vilmos Adleff[2], James R. White[2], Claudio Luchini [7], Antonio Pea[8], Aldo Scarpa [7,9], Giovanni Butturini[10], Giuseppe Zamboni [7,11], Paola Castelli[11], Seung-Mo Hong [1,12], Shinichi Yachida [13], Nobuyoshi Hiraoka [14], Anthony J. Gill [15,16,17], Jaswinder S. Samra[15,18,19], G. Johan A. Offerhaus[3], Anne Hoorens [20], Joanne Verheij[21], Casper Jansen[22], N. Volkan Adsay[23], Wei Jiang[24], Jordan Winter[25,26], Jorge Albores-Saavedra[27], Benoit Terris[28], Elizabeth D. Thompson[1], Nicholas J. Roberts [1,2], Ralph H. Hruban [1,2], Rachel Karchin [2,4,5],

Robert B. Scharpf[2], Lodewijk A. A. Brosens [3,29], Victor E. Velculescu [1,2,4] & Laura D. Wood [1,2✉]

[1]Department of Pathology, Sol Goldman Pancreatic Cancer Research Center, Johns Hopkins University School of Medicine, Baltimore, MD, USA. [2]Sidney Kimmel Comprehensive Cancer Center, Johns Hopkins University School of Medicine, Baltimore, MD, USA. [3]Department of Pathology, The University Medical Center Utrecht, Utrecht, The Netherlands. [4]Institute for Computational Medicine, Johns Hopkins University, Baltimore, MD, USA. [5]Department of Biomedical Engineering, Johns Hopkins University, Baltimore, MD, USA. [6]Department of Pathology and Molecular Diagnostics, Aichi Cancer Center, Nagoya, Japan. [7]Department of Diagnostics and Public Health, Section of Pathology, University of Verona, Verona, Italy. [8]Department of Surgery – The Pancreas Institute, University and Hospital Trust of Verona, Verona, Italy. [9]ARC-Net Centre for Applied Research on Cancer, University and Hospital Trust of Verona, Verona, Italy. [10]Department of Surgery, Pederzoli Hospital, Peschiera del Garda, Italy. [11]Pathology, IRCCS Sacro Cuore Don Calabria Hospital, Negrar, Verona, Italy. [12]Department of Pathology, Asan Medical Center, University of Ulsan College of Medicine, Seoul, Republic of Korea. [13]Department of Cancer Genome Informatics, Graduate School of Medicine, Osaka University, Osaka, Japan. [14]Department of Pathology and Clinical Laboratories, National Cancer Center Hospital, Tokyo, Japan. [15]University of Sydney, Sydney, NSW, Australia. [16]NSW Health Pathology, Department of Anatomical Pathology, Royal North Shore Hospital, Sydney, NSW, Australia. [17]Cancer Diagnosis and Pathology Research Group, Kolling Institute of Medical Research, Royal North Shore Hospital, Sydney, NSW, Australia. [18]Upper Gastrointestinal Surgical Unit, Royal North Shore Hospital, Sydney, NSW, Australia. [19]Faculty of Medical and Health Sciences, Macquarie University, Sydney, Australia. [20]Department of Pathology, Ghent University Hospital, Ghent, Belgium. [21]Department of Pathology, Academic Medical Center, Amsterdam, The Netherlands. [22]LABPON, Laboratory for Pathology Eastern Netherlands, Hengelo, The Netherlands. [23]Koc University School of Medicine, Istanbul, Turkey. [24]Department of Pathology, Thomas Jefferson University, Philadelphia, PA, USA. [25]University Hospitals Cleveland Medical Center and Seidman Cancer Center, Cleveland, OH, USA. [26]Case Comprehensive Cancer Center, Cleveland, OH, USA. [27]Department of Pathology, Medica Sur Clinic and Foundation, Mexico City, Mexico. [28]Service de Pathologie, AP-HP, Hôpital Cochin, Université Paris Descartes, Paris, France. [29]Department of Pathology, Radboud University Medical Center, Nijmegen, The Netherlands. ✉email: ldwood@jhmi.edu

