## [Peer Review File · Nature Communications]

Reviewers' comments:

Reviewer #1 (Remarks to the Author):

The authors claim that their data “established that both IPMNs and MCNs are direct precursors of invasive pancreatic cancer”. This would be the most important conclusion of this manuscript and, if true, it is a significant observation. An important limitation as mentioned by the authors is the small number of patients. However, as the authors point out, this is the largest study of that sort published to date.

The authors' central conclusion that IPMNs/MCNs are in lineage with invasive pancreatic cancer was argued only by point mutations and not rearrangements. Point mutation based lineage analysis suffers from two issues: 1. Some point mutations (such as commonly seen driver mutations i.e. KRAS_G12V) are not clonally specific as they can be shared among inter-patient samples and can appear in intra-patient samples by chance. 2. Even if LCM was used in some cases, there is a possibility from contamination from the invasive component as invasive cells migrate. Therefore, in this reviewer's opinion, it would be helpful to alleviate these two concerns by some additional analysis and clarification. For example, what would happen if all specific point mutations that are found in cancers (or maybe pancreatic cancers) across individuals such as the KRAS (G12V) would be eliminated from the analysis? And what if all mutations that are found in more than one patient in the current study are not used in the analysis? Statistical analysis that shows the likelihood that a patient-specific mutation can be seen in random in intra-patient samples could augment their conclusion. Allelic frequency analysis of shared mutations can also be used to eliminate the possibility of contamination.

The detection of many driver mutations in only pre-invasive neoplasms is also curious. It is true that subclones evolve on their own and could accumulate mutations that are different than the invasive component, but this is observed here in commonly seen driver genes in invasive pancreatic cancer. Furthermore, the authors found IPMN regions that shared no mutations with the invasive cancer, suggesting that they represented genetically independent clones. The authors also point out that there is a notably higher degree of intra-patient similarity between invasive and non-invasive samples in copy-number space than the mutational space but did not give any biological explanation. Taken together, the information presented by the authors makes their claim that “IPMNs and MCNs are direct precursors of invasive pancreatic cancer” somewhat questionable but not indisputable. The manuscript might improve if some of the points mentioned above are clarified.

The SMAD4 story is also perplexing and should be further clarified because, as presented in this manuscript, it will likely confuse the scientific community. SMAD4 alterations in both IPMNs and invasive cancers are found in this study. We know that in invasive carcinoma biallelic inactivation of SMAD4 is very common. This can happen by combinations of mutations and rearrangements. However, this study is not comprehensive as it lacks the SMAD4 alterations that are due to junctions or deletions. Therefore, the question of the possibility of SMAD4 biallelic inactivation in IPMN/MCNs has not been answered. Could it be possible that only monoallelic SMAD4 alterations are present in

IPMN/MCN and it is the second hit on the other allele that turns them to invasive? Also there is a possibility that the detected SMAD4 mutations in pre-invasive neoplasia samples are due to contamination. This point should be mentioned as a limitation or further be argued using allelic frequencies.

Reviewer #2 (Remarks to the Author):

In this manuscript, Noe and colleagues have conducted whole exome sequencing on paired IPMNs/MCNs and associated invasive cancers from 18 patients. In addition they have conducted targeted multi region sequencing from additional non invasive foci of IPMNs/MCNs. Using this data, they have constructed an evolutionary cascade of IPMN/MCN progression to invasive cancer, including putative timelines of progression.

The major findings of the study are as follows:

- A: IPMNs and MCNs are precursors of the associated pancreatic cancer.
- B: Non invasive IPMNs and MCNs in a single patient may contain heterogeneous driver mutations such as in KRAS or RNF43.
- C: Some IPMNs represent cancerization of the ducts by retrograde extension of the cancer.
- D: Some mutations in IPMNs/MCNs are early (KRAS, GNAS) and others are late (SMAD4, TGFBR2)
- E: IPMNs and MCNs do not progress overnight to pancreatic cancers and there is a timeline of progression in terms of several years. This represents a window of opportunity for early detection.

Each and every one of these findings has been demonstrated previously by work from Johns Hopkins and Memorial Sloan Kettering investigators in the field. While I have no quibbles with the volume and rigor of work, I am unable to grasp the novelty of the findings in this manuscript.

Minor point: I would be very cautious about 17% somatic mutation rate in ATM especially from archival material. ATM (a large gene) has a lot of dubious "mutations" in literature due to spurious variant calling.

RESPONSE TO REVIEWERS

We appreciate the thorough evaluation of our study by the reviewers and editors. In the revised version of our manuscript, we address essentially all of the points raised by the reviewers, and we think the resulting study is much improved. Specific revisions and responses to each comment are provided in detail below.

Reviewer #1 (Remarks to the Author):

The authors claim that their data "established that both IPMNs and MCNs are direct precursors of invasive pancreatic cancer". This would be the most important conclusion of this manuscript and, if true, it is a significant observation. An important limitation as mentioned by the authors is the small number of patients. However, as the authors point out, this is the largest study of that sort published to date.

Response: We thank the reviewer for the thoughtful review and comments on our study.

The authors' central conclusion that IPMNs/MCNs are in lineage with invasive pancreatic cancer was argued only by point mutations and not rearrangements. Point mutation based lineage analysis suffers from two issues: 1. Some point mutations (such as commonly seen driver mutations i.e. KRAS_G12V) are not clonally specific as they can be shared among inter-patient samples and can appear in intra-patient samples by chance. 2. Even if LCM was used in some cases, there is a possibility from contamination from the invasive component as invasive cells migrate. Therefore, in this reviewer's opinion, it would be helpful to alleviate these two concerns by some additional analysis and clarification. For example, what would happen if all specific point mutations that are found in cancers (or maybe pancreatic cancers) across individuals such as the KRAS (G12V) would be eliminated from the analysis? And what if all mutations that are found in more than one patient in the current study are not used in the analysis? Statistical analysis that shows the likelihood that a patient-specific mutation can be seen in random in intra-patient samples could augment their conclusion. Allelic frequency analysis of shared mutations can also be used to eliminate the possibility of contamination.

Response: We agree with the reviewer that somatic mutations in hotspot positions, such as codon 12 of *KRAS*, can confound analyses of relatedness between lesions. These hotspot mutations are a particular challenge in evolutionary analyses using targeted sequencing of small gene panels due to the limited number of mutations identified. However, in the current study, we employed whole exome sequencing, identifying an average of 66 somatic mutations per analyzed sample, the vast majority of which did not occur in hotspot positions. Moreover, an average of 47 somatic mutations were shared between IPMN/MCN and cancer samples, underscoring that our evolutionary analyses are based on a large number of mutations, including both driver and passenger alterations. Nevertheless, to address the reviewer's comment on the potential confounding effect of hotspot somatic mutations on our evolutionary analyses, we repeated the evolutionary analyses for each sample after excluding previously reported driver gene mutations (defined as inactivating mutations in tumor suppressor genes and nonsynonymous mutations in cancer-associated genes in which the identical amino acid change was observed at least 10 times in COSMIC) as well as alterations observed in more than one patient in the current study. The resulting phylogenies (generated using only passenger mutations) were essentially identical to those produced using all somatic mutations, demonstrating that hotspot mutations did not influence our analyses. The

most complex phylogenies are shown below, demonstrating minimal changes after exclusion of hotspot mutations. The remaining phylogenies had no change aside from adjustments to branch length due to the different numbers of analyzed mutations. Overall, the similarities observed in the new analyses are likely due to the large number of non-hotspot mutations identified by whole exome sequencing that were used in the original analyses. We have now included a statement in the Results section regarding the robustness of our evolutionary analyses even after the exclusion of hotspot mutations (page 6). In addition, as suggested by the reviewer, we have included a comment in the Results section on the extremely low probability of sharing a non-hotspot mutation due to chance alone (page 6).

We also agree with the reviewer that the presence of a small number of invasive carcinoma cells in the IPMN/MCN samples could result in misleading conclusions about evolutionary relationships. Our detailed pathological characterization and macro- or microdissection minimized the risk of this sample impurity. In addition, as the reviewer points out, variant allele frequencies (VAFs) of shared mutations in IPMN/MCN and cancer samples can also help to evaluate the likelihood of such contamination. Based on the reviewer's suggestion, we have now performed analyses of VAFs for all mutations in each sample and included these in Supplementary Figures 1-18. These analyses demonstrate that for mutations shared between the IPMNs/MCNs and cancers, the mean VAF in the IPMN/MCN samples was 0.38, with only 4% of shared mutations having a VAF below 0.1 in the IPMN/MCN samples. These high VAFs demonstrate that the shared mutations were not the result of a small number of cancer cells contaminating the IPMN/MCN samples. We have now included these analyses in the Results section of the revised manuscript (pages 6-7).

The detection of many driver mutations in only pre-invasive neoplasms is also curious. It is true that subclones evolve on their own and could accumulate mutations that are different than the invasive component, but this is observed here in commonly seen driver genes in invasive pancreatic cancer. Furthermore, the authors found IPMN regions that shared no mutations with the invasive cancer, suggesting that they represented genetically independent clones. The authors also point out that there is notably higher degree of intra-patient similarity between invasive and non-invasive samples in copy-number space than the mutational space but did not give any biological explanation. Taken together, the information presented by the authors makes their claim that "IPMNs and MCNs are direct precursors of invasive pancreatic cancer" somewhat questionable but not indisputable. The manuscript might improve if some of the points mentioned above are clarified.

Response: We agree with the reviewer that several of the findings in our study are unexpected, including the presence of driver gene mutations limited to IPMN/MCN samples and the existence of IPMN regions genetically unrelated to other regions within the same IPMN. Although most driver alterations are shared between IPMN/MCN and cancer samples (Figure 1b), the fact that some driver alterations are seen only in IPMN/MCN samples highlights the independent evolution of these lesions. Such independent evolution of precursor lesions has been observed in other settings (Labidi-Gray *et al.* Nat Commun 2017, Chen *et al.* Nat Commun 2017, Li *et al.* Gut 2019) and does not diminish the conclusion of our evolutionary analyses that IPMNs/MCNs are precursors of invasive pancreatic cancer. Nevertheless, we appreciate the reviewer highlighting these important points, and we present a full discussion of these findings along with their biological implications in the Discussion of the revised manuscript.

In addition, in reference to reviewer's point about comparing point mutations and copy number changes, we have now included the mean number of shared copy number alterations per sample (5), as well as the mean proportion of copy number alterations that are unique to cancer samples (0.34) in the Results section of the revised manuscript (pages 5-6). This proportion is similar to the mean proportion of point mutations unique to cancer samples (0.28). Thus, while the number of copy number alterations per sample is smaller than the number of point mutations, the proportion of alterations limited to the cancer is similar between both alteration types.

The SMAD4 story is also perplexing and should be further clarified because, as presented in this manuscript, it will likely confuse the scientific community. SMAD4 alterations in both IPMNs and invasive cancers are found in this study. We know that in invasive carcinoma biallelic inactivation of SMAD4 is very common. This can happen by combinations of mutations and rearrangements. However, this study is not comprehensive as it lacks the SMAD4 alterations that are due to junctions or deletions. Therefore, the question of the possibility of SMAD4 biallelic inactivation in IPMN/MCNs has not been answered. Could it be possible that only monoallelic SMAD4 alterations are present in IPMN/MCN and it is the second hit on the other allele that turns them to invasive? Also there is a possibility that the detected SMAD4 mutations in pre-invasive neoplasia samples are due to contamination. This point should be mentioned as a limitation or further be argued using allelic frequencies.

Response: We agree with the reviewer that the presence of *SMAD4* mutations in IPMN/MCN samples is unexpected and that additional data on the mono-allelic or bi-allelic nature of these mutations would be valuable. We have analyzed the *SMAD4* status in each sample using both somatic mutations and copy number alterations and have included this information in the revised manuscript. We now present data to show whether each mutated sample has mono-allelic or bi-allelic alteration of *SMAD4* in Supplementary Table 7. These analyses show that both precancer and cancer samples are enriched for bi-allelic *SMAD4* mutations (5/7 bi-allelic in precancer, 6/7 bi-allelic in cancer) and do not support the hypothesis that *SMAD4* alterations are largely mono-allelic in IPMNs/MCNs. In addition, the average VAF of *SMAD4* mutations in IPMN samples is 0.65, demonstrating that these mutations are not called due to contamination of a small number of *SMAD4*-mutant invasive carcinoma cells in otherwise *SMAD4*-wildtype IPMN/MCN samples. We also identify cases with IPMN/MCN-specific *SMAD4* mutations (samples MTP1, MTP3), further arguing against contamination with invasive carcinoma as the source of *SMAD4* mutations in IPMNs/MCNs. Finally, we agree that our whole exome sequencing analysis was not able to detect other types of *SMAD4* alterations, as alterations due to genomic rearrangements or epigenetic silencing were not detectable with our approach. We have added these points to the Results and Discussion of the revised manuscript (pages 9 and 13).

Reviewer #2 (Remarks to the Author):

In this manuscript, Noe and colleagues have conducted whole exome sequencing on paired IPMNs/MCNs and associated invasive cancers from 18 patients. In addition they have conducted targeted multi region sequencing from additional non-invasive foci of IPMNs/MCNs. Using this data, they have constructed an evolutionary cascade of IPMN/MCN progression to invasive cancer, including putative timelines of progression.

Response: We thank the reviewer for the thoughtful review and comments on our study.

The major findings of the study are as follows:

A: IPMNs and MCNs are precursors of the associated pancreatic cancer.

B: Non-invasive IPMNs and MCNs in a single patient may contain heterogeneous driver mutations such as in KRAS or RNF43.

C: Some IPMNs represent cancerization of the ducts by retrograde extension of the cancer.

D: Some mutations in IPMNs/MCNs are early (KRAS, GNAS) and others are late (SMAD4, TGFBR2)

E: IPMNs and MCNs do not progress overnight to pancreatic cancers and there is a timeline of progression in terms of several years. This represents a window of opportunity for early detection.

Each and every one of these findings has been demonstrated previously by work from Johns Hopkins and Memorial Sloan Kettering investigators in the field. While I have no quibbles with the volume and rigor of work, I am unable to grasp the novelty of the findings in this manuscript.

Response: We agree that investigators at Johns Hopkins (including our own group) and Memorial Sloan Kettering have previously performed molecular analyses of IPMNs. These efforts have included two studies that performed paired molecular analysis of IPMNs and associated carcinomas, as well as one additional study from investigators in Japan, all of which are cited in our study (Tan *et al.* J Am Coll Surg 2015; Felsenstein *et al.* Gut 2018; Omori *et al.* Gastroenterology 2019). However, all three of these studies performed targeted DNA sequencing of paired IPMN and cancer samples, analyzing only a limited number of genes (18-275 genes), rather than whole exome analyses (~20,000 genes). This is a major limitation of these previous studies, as they were unable to comprehensively assess alterations throughout the genome. While smaller targeted analyses often identify only a handful of alterations, typically in the same driver genes, a whole exome analysis can identify alterations in the compendium of human genes, including in both driver and passenger genes, permitting an unbiased evolutionary analysis of malignant progression. Highlighting this unbiased sequencing approach, we were able to identify multiple previously unreported driver genes in the IPMN pathway, including *GLI3* and *SF3B1*. Moreover, although other studies have analyzed these lesions, our study provides the first unbiased multiregion evolutionary analyses of IPMNs and associated carcinomas. We highlight how our study is novel compared to the specific reviewer's points indicated above and related publications:

A. Clinical and pathological features have previously suggested that IPMNs and MCNs are precursors of invasive pancreatic cancer, and previous studies have shown shared somatic mutations between adjacent non-invasive and invasive neoplasms (including Tan *et al.* J Am Coll Surg 2015; Felsenstein *et al.* Gut 2018; Omori *et al.* Gastroenterology 2019). However, our study represents the first evolutionary analysis to conclusively demonstrate the phylogenetic relationship between IPMNs/MCNs and associated invasive carcinomas, confirming their status as true premalignant lesions.

B. Though heterogeneity with respect to driver gene mutations in IPMNs has been shown by our group and others (Tan *et al.* J Am Coll Surg 2015; Felsenstein *et al.* Gut 2018; Kuboki *et al.* J Pathol 2018; Fischer *et al.* Gastroenterology 2018), the current study is the first whole exome analysis to demonstrate that these patterns of heterogeneity are unique to specific driver genes.

C. To our knowledge, this study is the first genetic confirmation of the process of "cancerization" mimicking IPMN, as none of the studies mentioned by the reviewer identified this process.

D. While the timing of driver gene mutations in IPMN tumorigenesis has been inferred by analysis of single samples from IPMNs with different grades of dysplasia (such as Amato *et al.* J Pathol 2014), our study shows the acquisition of driver gene mutations within individual lesions. In particular, we identify mutations in *SMAD4* and *TGFBR2* which are limited to invasive carcinoma in several cases. Though such cancer-specific mutations in *SMAD4* have been previously identified, our study represents the first to highlight them as unique across the exome as potential drivers of invasion. For example, although Tan *et al.* (J Am Coll Surg 2015) identified alterations of *SMAD4*, they do not mention this gene in the text or conclusions of their paper. In addition, our finding that *RNF43* mutations are enriched in non-invasive components is novel and suggests a unique and unexplored role for this gene in premalignant pancreatic tumorigenesis.

E. While we agree with the reviewer that some clinical data suggest that progression to carcinoma does not occur “overnight”, our study represents the first use of genomic data from precancerous lesions to estimate the timeline of this malignant progression. Surprisingly, other studies analyzing only cancer samples (without the benefit of an evolutionary analysis like ours) have even suggested that catastrophic genomic events can lead to rapid progression to pancreatic cancer (Notta *et al.* Nature 2016). Our study highlights the importance of such evolutionary timelines based on unbiased genomic analyses of precursor lesions, identifying a window of over 3 years for progression of IPMN to cancer and providing a foundation for early detection and surveillance approaches.

Overall, our work definitively demonstrates that IPMNs and MCNs are precursors of pancreatic cancer and provide new genomic insights in this disease. Nevertheless, in the light of the reviewer’s points raised above, we have included additional discussion of these aspects in the revised version of our manuscript (page 15).

Minor point: I would be very cautious about 17% somatic mutation rate in *ATM* especially from archival material. *ATM* (a large gene) has a lot of dubious "mutations" in literature due to spurious variant calling.

Response: We agree with the reviewer that *ATM* is a large gene and as such more likely to accumulate passenger or artifactual mutations than small genes. Still, our mutations are called using a rigorous and well-validated pipeline (published in Jones *et al.* Sci Transl Med 2015, Sausen *et al.* Nat Commun 2015, Le *et al.* N Engl J Med 2015, Anagnostou *et al.* Cancer Discov 2017, Labidi-Gray *et al.* Nat Commun 2017, Forde *et al.* N Engl J Med 2018, Papp *et al.* Cell Rep 2018, Anagnostou *et al.* Nat Cancer 2020, among others), which includes a VAF threshold, visual inspection, analyses of related regions in the genome using BLAT, and other bioinformatic steps to remove artifactual changes. Of note, larger genes than *ATM* (such as *TTN*), have a lower mutation prevalence, suggesting that at least some of the alterations called in *ATM* are true somatic mutations. Still, we have added a caveat to the Discussion section of the revised manuscript regarding the interpretation of *ATM* mutations in the context of its large gene size (page 12).

MTP1

CDKN2A:N39fs, KIF2B:A112T, KRAS:G12V, TP53:R337L

All Mutations

Passenger Mutations

MTP2

CDKN2A:Y44fs, ERBB4:R1067*, KRAS:G12V, RNF43:P231L, TP53:H214fs, TP53:P72R

All Mutations

Passenger Mutations

MTP3

CTNNB1:S45F, KRAS:G12D, PIK3CA:E545K, RNF43:A244fs, RNF43:D144fs, RNF43:E281*, RNF43:R132*, RNF43:R132*, RNF43:S121*, RNF43:S262*, SMAD4:E33fs, STK11:1192fs

All Mutations

Passenger Mutations

MTP5

ATM:D1853N, ATM:E958*, GNAS:R844C, KRAS:G12V, RNF43:P231L, RNF43:Q379*, SMAD4:K113fs, TGFB2:R553H

All Mutations

Passenger Mutations

MTP8

GNAS:R844H, KRAS:G12D, RNF43:C471fs, RNF43:E39fs, RNF43:M1_G4del
RNF43:P231L, RNF43:P660fs, RNF43:P671fs, RNF43:R371*, RNF43:S321*, TGFBR2:R559H

All Mutations

Passenger Mutations

MTP24

KRAS:G12D, KRAS:G12V, TET2:Q997*

All Mutations

Passenger Mutations

MTP19

GNAS:R844C, KRAS:G12D, KRAS:G12R, KRAS:G12V, KRAS:Q61H:12:25980275_T/A
KRAS:Q61H:12:25980275_T/G, SUPT6H:1104T, TP53:E171*, TP53:R175H

All Mutations

Passenger Mutations

REVIEWERS' COMMENTS:

Reviewer#1:

the authors addressed all my concerns adequately.

Reviewer#2:

I appreciate the extensive comments made by the authors made on exactly delineating why this study demonstrates novelty compared to prior data. Also the clarification on ATM is accepted.